# STINMatch: Semi-Supervised Semantic-Topological Iteration Network for Financial Risk Detection via News Label Diffusion

**Xurui Li**[1]*, **Yue Qin**[2]⋆, **Rui Zhu**[2], **Tianqianjing Lin**[3], **Yongming Fan**[4], **Yangyang Kang**[1]†,
**Kaisong Song**[5,1], **Fubang Zhao**[1], **Changlong Sun**[1], **Haixu Tang**[2], **Xiaozhong Liu**[6]†

[1] Alibaba Group, China, [2] Indiana University Bloomington, USA, [3] Zhejiang University, China,
[4] Purdue University, USA, [5] Northeastern University, China, [6] Worcester Polytechnic Institute, USA

xurui.lee@msn.com, qinyue@iu.edu, zhu11@iu.edu, lintqj@zju.edu.cn, fan322@purdue.edu, xliu14@wpi.edu
{yangyang.kangy,kaisong.sks,fubang.zfb}@alibaba-inc.com, changlong.scl@taobao.com, hatang@indiana.edu

## Abstract

Commercial news provide rich semantics and timely information for automated financial risk detection. However, unaffordable large-scale annotation as well as training data sparseness barrier the full exploitation of commercial news in risk detection. To address this problem, we propose a semi-supervised *Semantic-Topological Iteration Network*, STINMatch, along with a News-Enterprise Knowledge Graph (NEKG) to endorse the risk detection enhancement. The proposed model incorporates a label-correlation matrix and interactive consistency regularization techniques into the iterative joint learning framework of text and graph modules. The carefully designed framework takes full advantage of the labeled and unlabeled data as well as their interrelations, enabling deep label diffusion coordination between article-level semantics and label correlations following the topological structure. Extensive experiments demonstrate the superior effectiveness and generalization ability of STINMatch[1].

## 1 Introduction

Financial risk detection for enterprises is an essential task to assess and estimate the dynamic fragility of the market. Efforts need to be made to probe the vulnerable enterprises and enable timely preparedness. Traditional methods often treat each enterprise individually and leverage the official information or relevant structured data from government agencies to assess risk (Ozbayoglu et al., 2020). However, these official data are often biased and lagged, making it difficult to identify risks accurately and timely (Bi et al., 2022). Commercial news mining offers another effective perspective for financial risk detection owing to the massive and timely information embedded in news articles (Walker, 2016; Calomiris and Mamaysky, 2019; Li et al., 2022). Nevertheless, challenges are persisting on how to efficiently utilize news for detecting financial risks for enterprises.

One of the key issues is the multi-label diffusion problem. One news document may carry multiple risk labels (e.g., a '*debt risk*' can be accompanied by '*litigation threat*'), and conventional methods are difficult to handle the mutual influences among these labels given the rapid growth and variety of streaming media. Recently, deep learning methods have achieved great success in the field of natural language processing. Deep multi-label text classification (MLTC) methods can be applied to explore the label correlations (Liu et al., 2017; Yang et al., 2018). Unfortunately, pure text-based methods cannot handle risk diffusion in the business ecosystem. In order to address this problem, we propose a new model, semantic-topological iteration network (STIN), to estimate the '*risk diffusion*' on the established news-enterprise knowledge graph (NEKG). Unlike previous graph neural network (GNN) methods for text analysis (Yang et al., 2021b,a; Pang et al., 2022; Zhao et al., 2023), our STIN model focus on the multi-label-correlation guided text-graph joint learning, hoping to capture the dissemination for various types of financial risks following the NEKG topological structure.

Another challenge is the limited annotation data of financial news due to domain expert scarcity or expensive labor cost. Semi-supervised learning (SSL) is a common method for solving data scarcity problems. Many SSL methods based on entropy minimization, consistency regularization or generic regularization have been proposed for low-resource analysis scenarios (Berthelot et al., 2019, 2020; Sohn et al., 2020). However, few studies have been carried out on the semi-supervised integration for text-graph joint learning framework, which could improve the performance for both modules

---

\* These authors contributed equally to this work.
† Corresponding authors.
[1] https://github.com/curryli/Semi-Supervised-Financial-Risk-Detection.git

by leveraging the unlabeled data more efficiently in scenarios similar to our task.

The contributions are summarized as follows:

• We propose a pioneer semi-supervised text-graph joint learning framework STINMatch. It fully exploits the semantic information and topological association for risk diffusion with limited annotation data.

• A novel content-label-topology aggregation mechanism is further introduced during the iteration of STIN model to handle the multi-label diffusion issues on text-attributed graphs.

• We release an NEKG dataset annotated with multiple financial risks, which leverages real-world enterprise relatedness and news-enterprise associations for risk detection.

• Extensive experiments demonstrate the detection effectiveness of STINMatch and its good generalization ability on NEKG dataset, as well as other two public datasets.

## 2 Related Work

**Financial Risk Detection**. Classification and regression algorithms as well as time series forecasting have been widely used in financial risk detection (Ozbayoglu et al., 2020; Sezer et al., 2020). However, such methods mainly rely on historical, structured data from corporate or government agencies which lack up-to-date information. Recently, unstructured textual data, such as business management reports and financial news, are adopted for financial risk detection due to richer information and better timeliness (Peng and Yan, 2021; Li et al., 2022). However, such methods overlook the interactions between news and enterprises for risk diffusion by simply leveraging sentiment analysis on each isolated document. A recent work (Bi et al., 2022) leverages financial news as intermediaries between enterprises to exploit their interactions, but textual contents of the news are neglected.

**Label Diffusion**. Label correlations (Kurata et al., 2016; Yang et al., 2018; Zhang et al., 2021) are widely employed to improve model performance for MLTC tasks. GNN methods have also been used to deal with label diffusion issues for text analysis. Most of them apply GNN on the extracted word/entity-level knowledge graph (Yang et al., 2021b) or label co-occurrence graph (Pal et al., 2020) to enrich representation for each independent sample. Other GNN methods utilize text-attributed node relatedness (Kipf and Welling,

2017; Alkhereyf and Rambow, 2020) to enhance node representation, but the text representations are fixed during training. Some recent works are combining GNNs with text classifiers to take advantage of both topology and semantic modeling. For example, GLEM (Zhao et al., 2023) proposes a variational expectation maximization framework to alternatively updates the text and graph modules separately. Nevertheless, different from previous works, our STINMatch method focuses on the semi-supervised integration for text-graph joint learning framework, as well as the multi-label diffusion upon typologies for text-attributed GNN works.

**Consistency Regularization for SSL**. Consistency regularization is a popular SSL approach to constrain model predictions being invariant to input noise. MixMatch (Berthelot et al., 2019) applies data augmentation techniques and introduces a unified loss for unlabeled data that seamlessly reduces entropy while maintaining prediction consistency. The modified versions such as ReMixMatch (Berthelot et al., 2020) and UDA (Xie et al., 2020) both use weakly-augmented examples to generate artificial labels and enforce consistency against strongly-augmented examples. FixMatch (Sohn et al., 2020) is a simplified version of ReMixMatch and UDA, which combines the pseudo-labeling with consistency regularization while removing many specified components (e.g., training signal annealing and distribution alignment). However, all these methods focus on SSL within a single text or graph module respectively and could not be trivially adapted to our joint learning framework for text-attributed graphs. Other related work related to classification for subjective texts in different granularities include (Xiao et al., 2019; Moon et al., 2021; Song et al., 2023).

## 3 Preliminary

In this section, we introduce the task goal of STINMatch, detail the NEKG construction, and provide an intuitive example for financial risk diffusion.

**Semi-supervised Risk Diffusion**. Given a set of news $X$ and risk labels $Y \in \{0, 1\}^K$, the dataset $D = D_L \cup D_U$ contains $n$ labeled news $D_L = \{(x_i, y_i)|x_i \in X, y_i \in Y, i = 1, 2, \ldots, n\}$ and $m$ unlabeled news $D_U = \{x_i|i = 1, 2, \ldots, m\}$. The risk diffusion model aims to learn the mapping $F : X \rightarrow Y$ from news to multiple financial risks.

**NEKG Construction**. The news-enterprise knowledge base is formulated as a graph $G = (N, C, E)$,

where $N$ is a vertex set denoting news, $C$ is a vertex set denoting enterprises, and $E = \{e|e = (p, q), p \in C, q \in C\} \bigcup \{e|e = (p, q), p \in N, q \in C\}$ is an undirected edge set. Among $N$, each news in $D_L$ is annotated with $K$ binary risk labels, represented by a $K$-hot vector.

_Resource and Statistics_. Specifically, our NEKG contains 99,666 news nodes and 50,193 enterprise nodes. Each news node is initialized by its title and content, and each enterprise node is initialized by the company name. The edge between a news node and an enterprise node indicates that the enterprise is mentioned in the content of the news. The edges between different enterprises belong to five real-world relationship types: _subsidiary, investment, share-manager, share-investor_, and _share-legal-entity_. In total, NEKG contains 135,340 news-enterprise edges and 121,938 enterprise-enterprise edges.

_Annotation_. We sample $|D_L| = 15,000$ news from all news data, and $D_L$ is annotated by three domain experts. Each news can be identified as correlating to one or more financial risks from the following labels: _Bankruptcy, Liquidation, Business closure, Production halts, Debt, Corruption, Dispute, Counterfeit, Fraud_, and _Litigation_. The annotation standard is summarized through three preliminary rounds of annotation with 500 pieces of news. After the adjustment through preliminary rounds, each annotator labels $D_L$ independently in more than a month's time, and the annotation results achieve a 0.803 Fleiss's kappa.

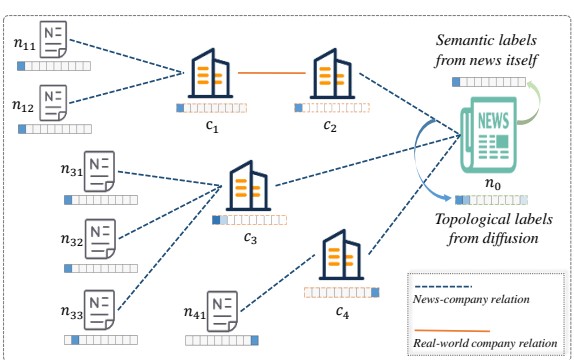

Figure 1: A toy example for financial risk diffusion through NEKG. NEKG consists of two node types: text-attributed news nodes and intermediate enterprise nodes.

**Intuitive Example**. Fig. 1 shows a toy example for explaining the financial risk diffusion through NEKG. The goal of model is to predict the risk label distribution of the target news node $n_0$. Tradi-

tional MLTC methods only consider the content of the news itself. Suppose the $n_0$ node are connected with three different enterprise nodes $c_{2,3,4}$ according to co-occurrence relations. Each enterprise node can be connected with other enterprise nodes according to the real-world enterprise relations (e.g. $[c_2, c_1]$), as well as some extra news nodes (e.g. $[c_3, n_{31}]$). The risk information from expanded news can be propagated to the center news node $n_0$ through the jump enterprise nodes. If most of the neighbor news contain the risk information of _Liquidation_, the probability for the target $n_0$ news having _Liquidation_ risk label also increases. On the other hand, the probability distribution of risk labels for each enterprise node can also be obtained when the diffusion model converges.

## 4  STINMatch: Methodology

### 4.1  Overview

**Architecture**. We propose an end-to-end semi-supervised semantic-topological iteration network to endorse multiple risk diffusion. As illustrated in Fig. 2, STINMatch is composed of a text classifier (i.e., $M_t$), a GNN-based node classifier (i.e., $M_g$), and a label-correlation matrix $\mathbb{R}$. $M_t$ contains a base text encoder and several classification layers. It takes textual inputs and learns text embeddings through classification objectives on labeled news. The hidden representations and predictions of $M_t$ are utilized to initialize node features of $M_g$ at each iteration round. $M_g$ propagates risk information from labeled samples to unlabeled samples through NEKG topology to achieve risk diffusion and boost risk detection. We iteratively train $M_t$ and $M_g$ in turn until convergence, and we seek to optimize the integration of the correlation matrix $\mathbb{R}$ into an iterative joint learning framework in order to maximize performance and efficiency. Upon the diffusion model reaching convergence, the enterprise risk labels are adopted as signals for precisely quantifying risk evaluation. Our work advocates for an innovative approach to financial risk detection that integrates label correlation effectively into the text-graph iterative learning process. The integration of label correlation into the iterative learning process offers a mutual benefit, significantly improving the overall performance of the system. On the one hand, the label correlation, serving as domain-specific knowledge, plays an important role to guide the label diffusion for both text and graph modules in each iteration; on the other

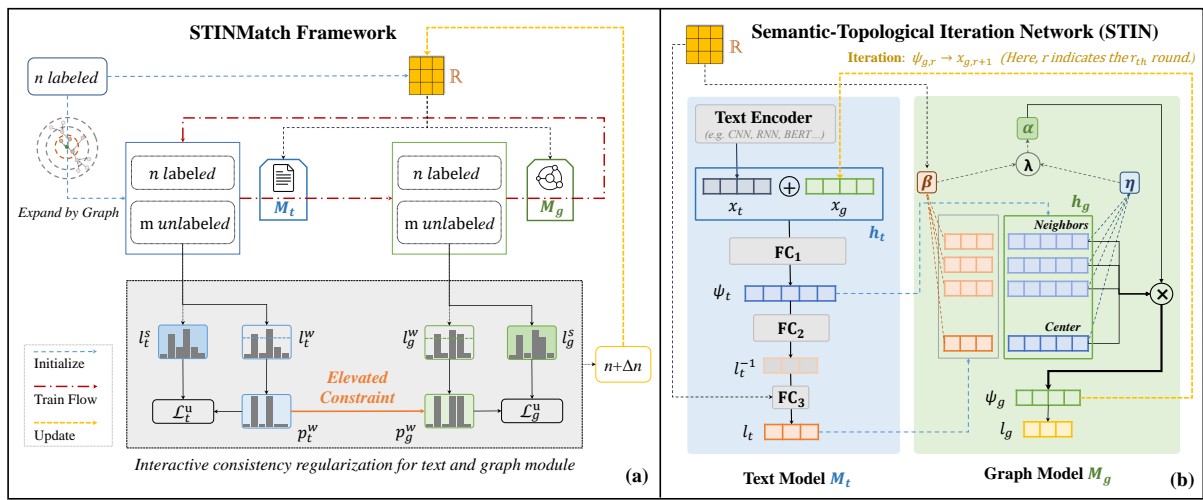

Figure 2: (a) shows the overall semi-supervised learning framework of STINMatch. (b) shows the deep interactions between text module, graph module and label correlations during the iteration for STIN model.

hand, the predictions of the successively enhanced model help involve more previously unlabeled samples into the calculation of label correlation, resulting in a more generalized label correlation in each iteration round.

**Iterative Learning Framework**. Below we elaborate on the iterative semi-supervised learning framework and the workflow from one iteration to the next iteration. We ignore the subscription denoting the iteration round for readability.

For a certain iteration round, we denote the key information layer in $M_t$ as $h_t$. $h_t$ carries two embeddings: text semantic embedding $x_t$, and graph context embedding $x_g$ from the last iteration. $M_t$ performs data augmentation techniques on both labeled and unlabeled data to compute supervised and unsupervised loss functions for updating model parameters. We detail the module components and semi-supervised learning strategy for $M_t$ in section 4.2.

After retraining $M_t$, we take the value of its representation layer $\psi_t$ as the learned text representation to initialize the node representations $h_g$ of $M_g$ in this iteration round. The $M_t$'s prediction $\ell_t := (p_1, p_2, \ldots, p_K)^T$ will also participate in the calculation of graph aggregation process. Similarly, we train $M_g$ with a joint loss function, consisting of a supervised loss and an unsupervised loss, and take the values of its last hidden layer $\psi_g$ as learned node representations. Then we use $\psi_g$ to reinitialize the $x_g$ of $M_t$ for the next iteration round. We detail the semi-supervised learning of $M_g$ and data filtering strategy in section 4.3.

With the retrained $M_t$ and $M_g$, STINMatch filters $\Delta n$ confidently-predicted samples from $m$ unlabeled samples to update the risk label correlation matrix $\mathbb{R} \in R^{K \times K}$ as $\mathbb{R}_{i,j} = \cos\langle (y_{1i}, y_{2i}, \ldots, y_{Ni}), (y_{1j}, y_{2j}, \ldots y_{Nj}) \rangle$, where $N = n + \Delta n$ is the number of samples for the joint set of $D_L$ and the filtered set, and $(y_{1i}, y_{2i}, \ldots, y_{Ni})$ is the vector consisting of the $i$-th risk labels for the $N$ samples. $\mathbb{R}$ is initially calculated from the labeled set $D_L$ and updated in each iteration round. Fig. 3 shows a visualized calculation process for $\mathbb{R}$.

## 4.2 Semi-Supervised Text Model

Text semantic embedding $x_t$ for $M_t$ comes from a text encoder (e.g. CNN, RNN or BERT-based). Graph context embedding $x_g$ is randomly initialized for all nodes and can be obtained from $\psi_g$ of graph model $M_g$ in the following iterations.

Similar to (Kurata et al., 2016), we involve a weight initialization strategy leveraging label co-occurrence to improve the model performance for MLTC task. Let $l_t^{-1}$ denote the second-last layer in $M_t$. The text model makes predictions as $l_t = (W_t \odot \mathbb{R}) l_t^{-1} + b_t$, where $\mathbb{R}$ is the risk label correlation matrix, $\odot$ represents element-wise production, and $W_t$ and $b_t$ are learnable parameters.

Let $\mathcal{X} = \{(x_b, y_b) : b \in (1, \ldots, B)\}$ denote a batch of labeled samples, where $x_b$ are training examples and $y_b$ are multi-hot labels. For the $k$-th independent risk label, the supervised loss on the batch of labeled data is:

$$\mathcal{L}_{t,k}^s = -\frac{1}{B}\sum_{i=1}^{B}[y_{i,k}\log p_{i,k} + (1-y_{i,k})\log(1-p_{i,k})].$$

Here $p_{i,k}$ is the probability of the $i$-th sample being predicted with the $k$-th risk label by $M_t$.

Let $\mathcal{U} = \{u_b : b \in (1,\ldots,B')\}$ denote a batch of unlabeled samples. STINMatch applies different augmentation techniques on the unsupervised data $\mathcal{U}$ to generate a set of weak augmentation data $\mathcal{U}_t^w$ and a set of strong augmentation data $\mathcal{U}_t^s$, by manipulating the primary semantic representation $x_t$ and the supplementary graph context representation $x_g$ for $M_t$. Let $l_t^w = (p_1^w, p_2^w, \ldots, p_K^w)^T$ and $l_t^s = (p_1^s, p_2^s, \ldots, p_K^s)^T$ denote the predictions of $M_t$ on $\mathcal{U}_t^w$ and $\mathcal{U}_t^s$, respectively. To generate $l_t^w$, the key idea is only to disturb the supplementary graph context embedding $x_g$ for weak augmentation. We apply the Random Perturbation method (Kumar et al., 2019) on $x_g$ and combine it with the original $x_t$ as the input of $FC_1$. For $l_i^s$, we applied a relative strong augmentation method Extrapolation (Kumar et al., 2019) on both $x_t$ and $x_g$, which utilizes the differences from other samples to synthesize new examples. Here, the weak augmentation provides higher accuracy for the pseudo labels, while strong augmentation provides better diversity and a larger region of sample perturbation for the consistency regularization, thereby improving the performance of the semi-supervised learning.

For each $p_k^w$, we calculate a pseudo-label with an indicator function $\hat{p}_k^w = \mathbb{1}[p_k^w > \tau]$ which returns 1 when $p_k^w > \tau$ else 0, where $\tau$ is a threshold hyperparameter. Let $P^w = \{\hat{p}_k^w | k = 1, 2, \ldots, K\}$. The unsupervised loss for $k$-th risk label on the batch of unlabeled data is defined as:

$$\mathcal{L}_{t,k}^u = -\frac{1}{B'}\sum_{i=1}^{B'}\mathbb{1}[\sum_{k=1}^{K}\hat{p}_{i,k}^w > 1]\cdot L_{t,k,i}^u,$$

where $\mathbb{1}[\sum_{k=1}^{K}\hat{p}_{i,k}^w > 1]$ is an indicator function for verifying the validity of the predictions on the $i$-th augmented sample and $L_{t,k,i}^u$ is the cross-entropy loss on $k$-th label for the $i$-th unlabeled sample:

$$L_{t,k,i}^u = [\hat{p}_{i,k}^w \log p_{i,k}^s + (1-\hat{p}_{i,k}^w)(1-\log p_{i,k}^s)].$$

Finally, we merge the supervised and unsupervised loss of all $K$ labels for training $M_t$: $\mathcal{L}_t = \sum_{i=1}^{K}\left[\mathcal{L}_{t,k}^s + \gamma\mathcal{L}_{t,k}^u\right]$, where $\gamma$ is a hyperparameter.

### 4.3 Semi-Supervised Graph Model

**Content-Label-Topology Aggregation**. The aggregation mechanism in $M_g$ is dominated by a *Semantic-representation and Label-distribution guided Attention* (SLA) layer. News semantics, label correlations, and risk diffusion can be learned jointly via NEKG by stacking SLA layers. The following part introduces the forward calculation from the input node feature set $h_g^l$ of the $l$-th SLA layer to that of the next layer $h_g^{l+1}$. Below we omit the subscription denoting the graph model $g$ and the number of layer $l$ for readability. For each node $\mu$, the input and output node features for SLA layer are denoted as $\vec{h_\mu}$ and $\vec{h'_\mu}$, respectively.

Specifically, we first apply a multi-head semantic-similarity-based attention mechanism similar to (Veličković et al., 2017). It learns $J$ independent semantic attention weights to stabilize the learning process, where the $j$-th single-head weight between node $\mu$ and $\nu$ is calculated as:

$$\eta_{\mu,\nu}^j = \text{LeakyReLU}(a^j[W^j\vec{h_\mu}||W^j\vec{h_\nu}]),$$

where $W^j$ is a shared transformation matrix, $a^j$ is a shared feed-forward neural network for each layer, and $||$ is the concatenation operation.

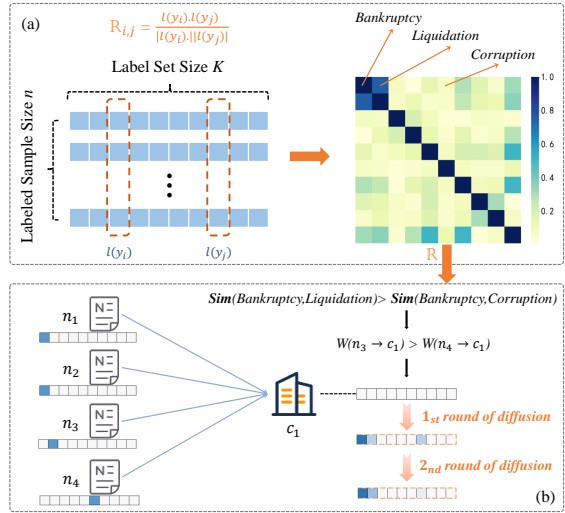

Figure 3: (a) shows the calculation process and a visualization of label correlation matrix $\mathbb{R}$. (b) is an example showing how does $\mathbb{R}$ affect the risk label diffusion.

Then we involve a label-similarity-based attention mechanism since there exist internal relations among different labels. However, direct similarity calculation between multi-hot label representations neglects such correlations. To address this issue, we utilize the correlation matrix $\mathbb{R}$ to capture the internal relations among different labels to enable cross-label similarity calculation. As shown in Fig. 3, without considering label correlations, the aggregation weight $W(n_3 \to c_1)$

is equal to $W(n_4 \rightarrow c_1)$. While considering that *Bankruptcy* and *Liquidation* are correlated risk labels, $W(n_3 \rightarrow c_1)$ becomes larger than $W(n_4 \rightarrow c_1)$. The label attention between node $\mu$ and node $\nu$ is:

$$\beta_{\mu,\nu} = ||\ell_{t,\mu} \cdot \ell_{t,\nu}^T \odot \mathbb{R}||_{\text{Frobenius}}.$$

Note that $\ell_{t,\mu}$ and $\ell_{t,\nu}^T$ come from the predictions of $M_t$, $\odot$ represents element-wise production, and $||.||_{\text{Frobenius}}$ represents frobenius norm for matrix.

After obtaining both $\eta_{\mu,\nu}$ and $\beta_{\mu,\nu}$, we combine them into a merged attention weight $\alpha$. The $j$-th merged attention part of node $\mu$ with respect to node $\nu$ is the softmax of a linear combination:

$$\alpha_{\mu,\nu}^j = \frac{\exp[\lambda^j \cdot \eta_{\mu,\nu}^j + (1 - \lambda^j) \cdot \beta_{\mu,\nu}]}{\Sigma_{r \in \mathcal{N}_\mu} \exp[\lambda^j \cdot \eta_{\mu,r}^j + (1 - \lambda^j) \cdot \beta_{\mu,r}]},$$

where $\mathcal{N}_\mu$ indicates the neighborhoods include itself for node $\mu$, and $\lambda \in R^J$ is a trainable vector. At last, the SLA layer outputs the feature representation for node $\mu$ as the concatenation of $J$ independent transformations:

$$\vec{h_\mu'} = \overset{J}{\underset{j=1}{\|}} \sigma \left( \sum_{\nu \in \mathcal{N}_\mu} \alpha_{\mu,\nu}^j \cdot W^j \vec{h_\nu} \right).$$

Here, $\sigma$ is the *sigmoid* function. Note that the input feature for the first SLA layer is initialized from the learned text representation $\psi_t$ in this iteration round, and we employ the averaging operation instead of concatenation for the $J$ head outputs on the final (prediction) layer as in (Veličković et al., 2017).

**Consistency Regularization for $M_g$.** Similar to $M_t$, the supervised loss for the $k$-th risk label to train $M_g$ on the batch of labeled data is:

$$\mathcal{L}_{g,k}^s = -\frac{1}{B} \sum_{i=1}^B [y_{i,k} \log q_{i,k} + (1 - y_{i,k}) \log(1 - q_{i,k})],$$

where $q_{i,k}$ is the probability for the $i$-th sample being predicted by $M_g$ as the $k$-th risk label.

Also, STINMatch applies data augmentations on the unsupervised data $\mathcal{U}$ to generate a set of weak augmentation data $\mathcal{U}_g^w$ and a set of strong augmentation data $\mathcal{U}_g^s$ for $M_g$. Let $l_g^w = (q_1^w, q_2^w, \dots, q_K^w)^T$ and $l_g^s = (q_1^s, q_2^s, \dots, q_K^s)^T$ denote the predictions of $M_g$ on $\mathcal{U}_g^w$ and $\mathcal{U}_g^s$, respectively. For $l_g^w$, we utilize the graph attention dropout on the attention coefficients in (Veličković et al., 2017) at inference phases. The weak augmentation only changes the aggregation pattern without disturbing the input node features. For strong augmented $l_g^s$, besides the graph attention

dropout, we further apply the Extrapolation augmentation method (Kumar et al., 2019) on the input node feature set $h_g$ of the graph model.

For each $q_k^w$, we also calculate a pseudo-label as $\hat{q}_k^w = \mathbb{1}[q_k^w > \tau]$, and let $Q^w = \{\hat{q}_k^w | k = 1, 2, \dots, K\}$. To ensure the validity of the predictions from $M_g$, we introduce an *Elevated Constraint* by restricting that the additional information from neighbors do not reduce the risk labels obtained from text model $M_t$ for each node itself. Namely only the samples whose pseudo-labels predicted from $M_t$ being a subset of that from graph module $M_g$ will participate in loss calculation, and the unsupervised loss for the $k$-th risk label to train $M_g$ on the batch of unlabeled data is defined as:

$$\mathcal{L}_{g,k}^u = -\frac{1}{B'} \sum_{i=1}^{B'} \mathbb{1} \left[ (\sum_{k=1}^K \hat{q}_{i,k}^w > 1) \wedge (P^w \subseteq Q^w) \right] \cdot L_{g,k,i}^u.$$

Here $\wedge$ represents the simultaneous satisfaction for both conditions. $L_{g,k,i}^u$ is the cross-entropy loss for the $i$-th unlabeled sample, regarding $\hat{q}_k^w$ as the label and $q_k^s$ as the prediction:

$$L_{g,k,i}^u = [\hat{q}_{i,k}^w \log q_{i,k}^s + (1 - \hat{q}_{i,k}^w)(1 - \log q_{i,k}^s)].$$

Finally, we merge the supervised and unsupervised loss of all $K$ labels for training $M_g$: $\mathcal{L}_g = \sum_{i=1}^K \left[ \mathcal{L}_{g,k}^s + \gamma \mathcal{L}_{g,k}^u \right]$.

# 5 Experiments

## 5.1 Experimental Setting

**Datasets.** We validate our model on three different datasets. The main dataset NEKG are described in section 3. Note that we fixed a 5000 original labeled news as test set. We take only a random part of the remaining original labeled news as our labeled data set $D_L$ for each experimental setting. We remove the labels except for $D_L$ during training STINMatch. The default labeled size for $D_L$ is set to 1000, unless otherwise stated. The enterprise-related information comes from a subset of the data integrated by our data center team collected from various of government-backed open data providers such as National Enterprise Credit Infomation Public System of China, and it enables the designed web-crawler to collect the corresponding news for annotations and evaluations.

The other two public datasets RentTheRunWay and Goodreads-Spoiler will be described in the algorithm generalization part of section 5.4.

**Implementation** We carried out all models with Pytorch. Graph model is implemented using dis-

tributed graphics library (DGL). All models are trained on the NVIDIA Tesla A100 80GB GPU. The hyper-parameter details are shown in Appendix A.

## 5.2 Baselines

As shown in Table 1, the STINMatch model is compared with baselines from 4 main categories.

**Category 1** contains classical text classification methods. RoBERTa-sw represents the RoBERTa trained using the sliding-window method (Wang et al., 2019). In order to validate the performance of different frameworks over pure text level fairly, most of the text encoder part in following categories are based on the TextCNN over pre-trained BERT embedding layers (represented by TCB).

**Category 2** contains state-of-the-art text models specialized for MLTC tasks such as LCNNI (Kurata et al., 2016), SGM (Yang et al., 2018) and CORE (Zhang et al., 2021). Focal or DB loss (Huang et al., 2021) designed to handle the label distribution for MLTC tasks from loss terms are also tested.

**Category 3** contains typical semi-supervised learning methods for pure text models.

**Category 4** applies typical GNN-based methods over TCB representations. General GNN models such as GCN (Kipf and Welling, 2017), GraphSAGE (Hamilton et al., 2017), GAT (Veličković et al., 2017) are used. GNN models specialized for MLTC tasks such as MAGNET (Pal et al., 2020) and LC-GAT (Xu et al., 2020) are also included for comparison. GLEM (Zhao et al., 2023) is a latest text-graph co-training method, which has also been tested. Some details of GLEM have been modified to adapt to our scenario for fair comparison (e.g. using TCB as base language module, replacing the binary cross-entropy loss with a multi-label loss).

## 5.3 Effectiveness for STINMatch

**News Risk Evaluation**. Experiment results are reported in Table 1 with the labeled size $n$ set to 1000. From Category 2 of Table 1 one can witness slight improvements with some multi-label methods, indicating the usefulness of correlations among labels. Text-based semi-supervised methods also slightly enhance the performances. GNN methods achieve certain improvements compared to that of pure text-based methods, showing the effectiveness of the additional NEKG. The proposed STINMatch model shows convincing superiority

Table 1: Comparisons on news risk detection task between our STINMatch model and other methods. † indicates that a pre-trained BERT embedding layer serves as the first layer of the model.

| Category | Model | MacroF1 | MicroF1 |
|---|---|---|---|
| Category 1 | FastText | 0.62 | 0.81 |
| | TextRCNN | 0.728 | 0.906 |
| | TextRCNN† | 0.748 | 0.885 |
| | TextRNN-attention† | 0.746 | 0.863 |
| | TextCNN† (i.e. TCB) | 0.808 | 0.922 |
| | RoBERTa-sw | 0.815 | 0.902 |
| Category 2 | LCNNI† | 0.724 | 0.831 |
| | SGM | 0.714 | 0.841 |
| | CORE† | 0.834 | 0.920 |
| | TCB-Focal-loss | 0.830 | 0.923 |
| | TCB-DB-loss | 0.813 | 0.917 |
| Category 3 | Pseudo-TCB | 0.814 | 0.921 |
| | MixMatch-TCB | 0.818 | 0.919 |
| | ReMixMatch-TCB | 0.821 | 0.921 |
| | FixMatch-TCB | 0.821 | 0.923 |
| Category 4 | GCN-TCB | 0.811 | 0.908 |
| | GraphSAGE-TCB | 0.827 | 0.916 |
| | GAT-TCB | 0.831 | 0.921 |
| | MAGNET-TCB | 0.835 | 0.923 |
| | LC-GAT-TCB | 0.837 | 0.926 |
| | GLEM-TCB | 0.849 | 0.927 |
| **STINMatch** | **STINMatch** | **0.889** | **0.938** |

Table 2: Evaluation for enterprise risk detection based on risk propagation results from different GNN models.

| Model | Accuracy | F1-low-risk | F1-high-risk |
|---|---|---|---|
| GCN-TCB | 0.668 | 0.786 | 0.264 |
| GraphSAGE-TCB | 0.677 | 0.792 | 0.270 |
| GAT-TCB | 0.695 | 0.807 | 0.281 |
| MAGNET-TCB | 0.684 | 0.798 | 0.274 |
| LC-GAT-TCB | 0.704 | 0.813 | 0.287 |
| GLEM-TCB | 0.716 | 0.824 | 0.295 |
| **STINMatch** | **0.732** | **0.834** | **0.310** |

compared to all baselines. It is owing to the multi-label-correlation guided text-graph joint learning, as well as the interactive semi-supervised learning across text-graph models. Note that we run the experiments for 5 times and reported the average performances. The t-test results show that the proposed model significantly outperforms the best baseline by 4.7%. The detailed ablation study results for STINMatch are described in Section 5.4.

**Enterprise Risk Evaluation**. We also conducted comparative experiments for evaluating enterprise risk detection based on the label-diffusion results from NEKG. In fact, each enterprise node has a credit risk rating label provided by an authoritative rating agency. The credit risk rating labels

can be divided into 2 different levels according to the rating scores: high-risk (1) and low-risk (including those with no risk) (0). On the other hand, the GNN-based models will also offer risk predictions for each enterprise node when converged. To make the two label systems comparable, we suppose enterprise nodes with more than one predicted risk labels from graph models are high-risk (1), while others are low-risk (0). We take the credit risk rating labels as ground truth, and calculate the classifier metrics (Accuracy and F1 for low-risk/high-risk class) according to the predictions from different GNN-based models.

By comparing the results of Table 1 and 2, we can find that the model performance of the enterprise risk classifier is nearly positively correlated with that on MLTC task for news risk detection. The performance of STINMatch method exceeds all that from other GNN-based baselines on enterprise risk evaluation task.

## 5.4 Analysis for STINMatch

**Ablation Study**. To investigate the contribution for each component in STINMatch, we conducted 6 different ablation studies. The first study was trained by patient epochs for text and graph module respectively with early stopping in a single round without iteration. The second study only initialized the label correlation matrix $\mathbb{R}$ once without the updating mechanism. The third study only considered the traditional semantic attention and ignored the label attention $\beta$ for $M_g$. The following studies explored different components for the semi-supervised learning.

From Table 3 we can find that every intentionally removed component leads to a decrease in the performance of the STINMatch model. It indicates that the well-trained semantic representation and graph-based diffusion model can perceive important reciprocal advantages from each other by leveraging the multi-label correlations. The two descriptions "w/o unsupervised loss" and "w/o unsupervised loss" indicate the results after removing the semi-supervised techniques from the text model and the graph model. It indicate that the interactive semi-supervised learning across text and graph modules indeed make a better utilization for unlabeled data at different training stages. Moreover, the superiority for the text-graph joint learning can be indicated by compare the results of category 3 from Table 1, which represent the semi-supervised-

Table 3: Ablation results for STINMatch method. w/o means without using corresponding component.

| Model | MacroF1 | MicroF1 |
|---|---|---|
| w/o iteration | 0.841 | 0.923 |
| w/o updating matrix $\mathbb{R}$ | 0.876 | 0.931 |
| w/o label attention $\beta$ | 0.869 | 0.925 |
| w/o unsupervised loss for $M_t$ | 0.873 | 0.929 |
| w/o unsupervised loss for $M_g$ | 0.871 | 0.929 |
| w/o elevated constraint | 0.878 | 0.931 |
| **STINMatch** | **0.889** | **0.938** |

enhanced text classification methods without text-graph joint learning.

**Time Cost** & **Labeled Size Analysis**. Fig. 4 presents the model performance over the increase with training epochs and the size of labeled data. The left insets (i.e., (a) and (b)) show the time cost and Macro F1 when varying training epochs with the labeled size fixed as 1,000. With nearly three times of computation time cost, our method outperforms the TCB baseline by 9.5% on Macro F1. The right insets show the increase ratio of Macro F1 along with the labeled data size. The result of inset (c) is calculated from dividing the best Macro F1 of STINMatch by that of TCB, which are shown in inset (d). It can be found that the increase ratio is larger when the labeled size is smaller, while it decreases with the growth of the labeled size. It indicates the better superiority of STINMatch in lower resource scenarios. It can also be observed from inset (d) that when the labeled size $|D_L|$ is 1,000, STINMatch achieves a performance which is approximately similar to that with $|D_L| = 10,000$. However, for the baseline TCB model, the performance of the model with $|D_L| = 1,000$ is significantly lower than that with $|D_L| = 10,000$. This indicates that the proposed mechanisms effectively reduce the dependency on labeled data, resulting in substantial cost savings in terms of annotation.

**Algorithm Generalization** We also processed two public datasets RentTheRunWay (Misra et al., 2018) and Goodreads-Spoilers (Wan et al., 2019) to demonstrate the generalization ability of our STINMatch method when adapted to other domains. We elaborate on the two review datasets in Appendix C.

For these two datasets, we fixed half of the samples as test set, and randomly selected 10% of the remaining samples as labeled ones for semi-supervised learning comparison. We selected four typical baselines described in section 5.2. From Ta-

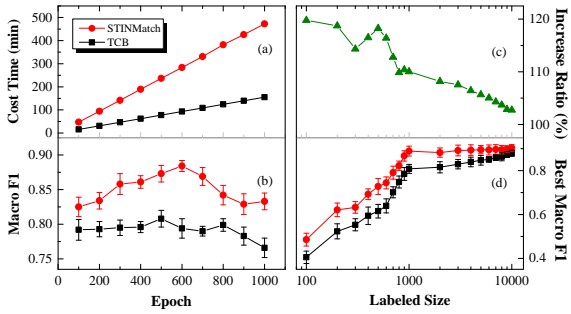

Figure 4: The left insets (a) and (b) show the variation trend of the time cost and model performance with the increase of training epochs. The right insets (c) and (d) show the variation trend on the best Macro F1 with increasing the labeled data size.

Table 4: Algorithm generalization experiments.

| Method | RentTheRunWay | | Goodreads-Spoilers | |
|---|---|---|---|---|
| | MacroF1 | MicroF1 | MacroF1 | MicroF1 |
| RoBERTa | 0.481 | 0.764 | 0.573 | 0.611 |
| GAT-TCB | 0.487 | 0.771 | 0.579 | 0.613 |
| GLEM-TCB | 0.501 | 0.789 | 0.585 | 0.624 |
| FixMatch-TCB | 0.495 | 0.786 | 0.584 | 0.621 |
| **SSINMatch-TCB** | **0.516** | **0.801** | **0.615** | **0.638** |

ble 4 we can see that our method still outperforms other baselines on the two public datasets, showing good generalization ability of STINMatch.

## 6 Conclusion

In this paper, we introduce the NEKG for helping detect financial risks from commercial news. The proposed STINMatch method outperforms existing state-of-the-art models on news risk detection task, as well as the downstream enterprise risk evaluation task. Such improvements mainly come from 1) Additional NEKG topology enables article-level risk diffusion; 2) The carefully designed STIN model brings deep interactions among semantic module, topological module and label-correlation matrix, enhancing the sound diffusion upon NEKG. 3) The innovative semi-supervised joint learning framework enables text and graph modules to perceive important reciprocal advantages from each other, making both modules utilize unlabeled data more effectively. The utility of real-world applications in scenarios similar to our financial risk detection task is substantial, ranging from multi-label document classification based on citation or social networks, to multi-label product classification through e-Commerce platforms. We also apply the STINMatch method on two public graph-enhanced

MLTC datasets and validate its good generalization ability.

## 7 Limitations

STINMatch leverages semi-supervised learning techniques on a semantic-topological iteration network. Our research demonstrates that both the joint iterative learning and interactive consistency regularization of the text and graph models benefit the risk diffusion. However, we acknowledge that the proposed mechanisms increase the computation cost. For example, with the same training epochs and batch size, the computation cost of STINMatch is nearly three times the simplest baseline (i.e., TCB). However, this higher cost is justified by the significant benefits it brings. In fact, STINMatch outperforms TCB by 9.5% in terms of Macro F1, which is a substantial improvement. Moreover, in order for TCB to achieve an evenly-matched performance, it needs almost 10 times labeled data than STINMatch. The overall results indicate that our STINMatch offers better results at a certain cost of consumption time.

## Acknowledgements

This work is supported by the National Natural Science Foundation of China (62106039).

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

## A  Hyperparameters

For all experiments, hyper-parameters are determined by grid search, and total maximum epochs are set to the same order with early stopping to be fair. The learning rate is is 2e-4 for the text model and 5e-4 for the graph model. The batch size $B$ is 8, $B'$ is 7 times greater than $B$, which is the same to (Sohn et al., 2020). The layers for graph model is 3. The confidence threshold parameter $\tau$ is set to 0.95. The balance weight parameter $\gamma$ for unsupervised and supervised loss is set to 0.5, the details can be seen in Table 5. The max content length in review datasets is 512, and the max length for each news in NEKG is set to 2000. For most experimental settings, we use a CNN-based text encoder with a pre-trained BERT embedding layer considering the length limitation and efficiency issues for fair comparison. Each text-graph iteration includes 100 text epochs and 100 graph epochs, the maximum of iterations is set to 10. All indicators are taken from the average value of last five epochs to ensure stability.

## B  NEKG Datasets

The distribution of each risk label among all annotated news the details can be seen in Table 6. Besides, we also report the circumstances of multi-label annotation of the news data in Table 7. Among all 15,000 annotated news, 5855 news are associated with 1 risk label, 6986 news are associated with 2 risk labels, and more than two thousand news have 3 or more labels.

## C  Review Datasets

RentTheRunWay (Misra et al., 2018) dataset contains self-reported fit feedback from customers as

Table 5: Performance under the balance weight parameter $\gamma$ at different scales for NEKG dataset.

| $\gamma$ | MacroF1 |
|---|---|
| 0.05 | 0.878 |
| 0.1 | 0.882 |
| 0.5 | 0.889 |
| 1 | 0.883 |

Table 6: Distribution of risk labels for NEKG dataset.

| Risk Label | Number of News |
|---|---|
| Bankruptcy | 3376 |
| Liquidation | 2107 |
| Business closure | 272 |
| Production halts | 705 |
| Debt | 4097 |
| Corruption | 474 |
| Dispute | 5262 |
| Counterfeit | 1187 |
| Fraud | 993 |
| Litigation | 8277 |

Table 7: Risk label count for news.

| Risk Label Count | Number of news |
|---|---|
| 1 | 5855 |
| 2 | 6986 |
| 3 | 1725 |
| 4 | 344 |
| 5 | 83 |
| 6 | 7 |

well as other side information like reviews, ratings, product categories, etc. It contains 105,508 users, 5,850 items and 192,544 reviews. We construct a bipartite graph of item nodes and review nodes. Item nodes are medium nodes similar to the enterprise nodes in our enterprise graph, and the review nodes are text-attributed nodes similar to our news nodes. One item is connected to another item by an edge if they are commented by a same user. We transformed the three-class fit-feedback tags and ten-class rating tags into the multi-labels.

Goodreads-Spoilers (Wan et al., 2019) dataset contains reviews from the Goodreads book review website. It contains 25,475 items, 18,892 users and 1,378,033 reviews. The constructing method of graph is similar to that of RentTheRunWay. We transformed the six-class rating tags and the "has-spoiler" tag into the multi-labels.

## D  Annotation Table

Finally, we provide a detailed annotation table for interpreting all symbols in Table 8.

Table 8: Annotation Table.

| Risk Label Count | Number of news |
|---|---|
| $\mathbb{R}$ | label correlation matrix |
| $K$ | number of all risk labels |
| $N$ | vertex set for news |
| $C$ | vertex set for enterprises |
| $E$ | edge set of news-enterprise KG |
| $D_L$ | labeled news data |
| $D_U$ | unlabeled news data |
| $M_g$ | graph module |
| $M_t$ | text module |
| $x_t$ | output of the text encoder in $M_t$ |
| $\psi_t$ | hidden representation in $M_t$ |
| $l_t$ | output of $M_t$ |
| $l_t^s$ | predictions on strong augmentation data in $M_t$ |
| $l_t^w$ | predictions on weak augmentation data in $M_t$ |
| $h_g$ | node representation in $M_g$ |
| $\psi_g$ | hidden representation in $M_g$ |
| $l_g$ | output of $M_g$ |
| $l_g^s$ | predictions on strong augmentation data in $M_g$ |
| $l_g^w$ | predictions on weak augmentation data in $M_g$ |
| $\beta$ | node-level attention weights regarding label correlation |
| $\eta$ | similarity-based node-level multi-head attention weights |
| $\alpha$ | node-level attention weights in $M_g$ |
| $\gamma$ | balance weight parameter for unsupervised and supervised loss |