# OpenReview forum: "STINMatch: Semi-Supervised Semantic-Topological Iteration Network for Financial Risk Detection via News Label Diffusion"
_EMNLP/2023/Conference — EMNLP 2023 Main_

### Official Review · Reviewer_CCXt · 2023-08-03

**Soundness:** 3

**Excitement:**

3: Ambivalent: It has merits (e.g., it reports state-of-the-art results, the idea is nice), but there are key weaknesses (e.g., it describes incremental work), and it can significantly benefit from another round of revision. However, I won't object to accepting it if my co-reviewers champion it.

**Missing References:**

N/A

**Paper Topic And Main Contributions:**

This paper proposes a novel semi-supervised method to fully leverage commercial news for automated financial risk. Specifically, it designs a semantic-topological iteration network (STIN) for multi-label risk diffusion, which mainly contains a content-label-topology aggregation mechanism to capture the interaction of STIN. Then, it integrates a semi-supervised learning method for a text-graph joint learning framework to address the limited annotation data of financial news. In addition, this paper releases a new dataset news-enterprise knowledge graph (NEKG) for multi-label financial risks. Results on real-world datasets show that this method achieves better performance than baselines.

**Questions For The Authors:**

a)   In section 3, the authors denote two edges in NEKG, i.e., news-news and news-enterprise E = {e|e = (p, q), p ∈ N, q ∈ N}∪ {e|e = (p, q), p ∈ N, q ∈C}. However, there is no news-news edge in the following sections but the enterprise-enterprise edge, for example, in Figure 1. Which one is correct?

b) The model framework in Figure 2 is hard to understand. Could you provide a necessary description of important symbols, such as M_t and M_g?


**Reasons To Accept:**

a)	This paper proposes a new financial risk detection method that could not only jointly leverage semantics of news and topological correlations to conduct better risk diffusion estimation but also could alleviate the limited annotation data problem by semi-supervised learning.

b)	This paper construct and release a new dataset NEKG with multiple financial risks, which could be helpful to other researchers.

c)	Extensive experiments on the constructed NEKG dataset and two public datasets show the effectiveness of this paper.


**Reasons To Reject:**

a)	This paper doesn’t analyze the influence of important hyperparameters, such as the balance weight parameter γ for unsupervised and supervised loss.

b)	This paper doesn’t introduce the risk label distribution in datasets, which is important to understand the experimental results.

c)	The model framework in Figure 2 is hard to understand. There should be a necessary description of important symbols.

d)	There are still some typos. In section 3, the authors denote two edges in NEKG, i.e., news-news and news-enterprise E = {e|e = (p, q), p ∈ N, q ∈ N}∪ {e|e = (p, q), p ∈ N, q ∈C}. However, there is no news-news edge in the following sections but the enterprise-enterprise edge, for example, in Figure 1.

e)	Some symbols are used but not defined. For example, K is not clearly introduced in the Semi-supervised Risk Diffusion module.


**Reproducibility:**

3: Could reproduce the results with some difficulty. The settings of parameters are underspecified or subjectively determined; the training/evaluation data are not widely available.

**Reviewer Confidence:**

3: Pretty sure, but there's a chance I missed something. Although I have a good feel for this area in general, I did not carefully check the paper's details, e.g., the math, experimental design, or novelty.

**Typos Grammar Style And Presentation Improvements:**

a)	There are still some typos. In section 3, the authors denote two edges in NEKG, i.e., news-news and news- enterprise E = {e|e = (p, q), p ∈ N, q ∈ N}∪ {e|e = (p, q), p ∈ N, q ∈C}. However, there is no news-news edge in the following sections but the enterprise- enterprise edge, for example, in Figure 1.

b)	Some symbols are used but not defined. For example, K is not clearly introduced in the Semi-supervised Risk Diffusion module.

---

> ### Author Rebuttal · Authors · 2023-08-28
>
> We are grateful for your insightful review comments, and we heartily welcome the opportunity to further articulate our position in explanation to the queries you raised.
>
> **Response to Question A**(*In section 3, the authors denote two edges in NEKG, i.e., news-news and news-enterprise E = {e|e = (p, q), p ∈ N, q ∈ N}∪ {e|e = (p, q), p ∈ N, q ∈C}. However, there is no news-news edge in the following sections but the enterprise-enterprise edge, for example, in Figure 1. Which one is correct?  *):
>
> - Thank you for pointing out the typo. The edge set should be E = {e|e = (p, q), p ∈ C, q ∈ C}∪ {e|e = (p, q), p ∈ N, q ∈C}.
>
> **Response to Question B**(*The model framework in Figure 2 is hard to understand. Could you provide a necessary description of important symbols, such as M_t and M_g?  *):
> - Thank you for the suggestions. We provide a notation table to describe all important symbols. Please refer to the response to  *Explanation of symbols* in *Response to "Reasons To Reject"*.
>
>
> **Response to "Reasons To Reject"**:
>  - **Influence of important hyperparameters**
>     - Thanks for your suggestions. Below we show the model performance under the balance weight parameter at different scales to explain our selection:
>
>     |**γ value**|**MacroF1**|
>     |-------------|--------------|
>     | 0.05	       | 0.878        |
>     | 0.1	        | 0.882        |
>     | 0.5	        | 0.889        |
>     | 1	          | 0.883        |
>     - We will supplement this information in the next version.
>
> - **Distribution of risk label**
>     - Thank you for pointing this out. Below we show the distribution of each risk label among all 15,000 annotated news.
>
>     |**Risk Label**    |**Number of news**|
>     | ----------- | ----------- |
>     |Bankruptcy|	3376|
>     |Liquidation|2107|
>     |Business closure|	272|
>     |Production halts|	705|
>     |Debt|	4097|
>     |Corruption|	474|
>     |Dispute|	5262|
>     |Counterfeit|	1187|
>     |Fraud	|993|
>     |Litigation|	8277|
>
>
>     - Besides, we report the circumstances of multi-label annotate for these news. We will supplement this information in the next version.
>
> |**Risk Label Count**|**Number of news**|
> |-----------------------| ----------- |
> | 1                     |	5855|
> | 2                     |	6986|
> | 3                     |	1725|
> | 4                     |	344|
> | 5                     |	83|
>
>  - **Explanation of symbols**
>
>     Following your suggestions, we provide a detailed annotation table for interpreting all symbols. We will supplement this information in the next version.
>
>     | Symbol     | Description                                                   |
>     | ----------- |---------------------------------------------------------------|
>     |$\mathbb R$| label correlation matrix                                      |
>     |$K$| 	number of all risk labels                                    |
>     |$N$| vertex set for news                                           |
>     |$C$| vertex set for enterprises                                    |
>     |$E$| edge set of news-enterprise KG                                |
>     |$D_L$| labeled news data                                             |
>     |$D_U$| unlabeled news data                                           |
>     |$M_g$| graph module                                                  |
>     |$M_t$| text module                                                   |
>     |$x_t$| output of the text encoder in $M_t$                           |
>     |$\psi_t$| hidden representation in $M_t$                                |
>     |$l_t$| output of $M_t$                                               |
>     |$l_t^s$| predictions on strong augmentation data in $M_t$              |
>     |$l_t^w$| predictions on weak augmentation data in $M_t$                |
>     |$h_g$| node representation in $M_g$                                  |
>     |$\psi_g$| hidden representation in $M_g$                                |
>     |$l_g$| output of $M_g$                                               |
>     |$l_g^s$| predictions on strong augmentation data in $M_g$              |
>     |$l_g^w$| predictions on weak augmentation data in $M_g$                |
>     |$\beta$| node-level attention weights regarding label correlation      |
>     |$\eta$| similarity-based node-level multi-head attention weights      |
>     |$\alpha$| node-level attention weights in $M_g$                         |
>     |$\gamma$| balance weight parameter for unsupervised and supervised loss |

---

### Official Review · Reviewer_BEjE · 2023-08-05

**Soundness:** 4

**Excitement:**

3: Ambivalent: It has merits (e.g., it reports state-of-the-art results, the idea is nice), but there are key weaknesses (e.g., it describes incremental work), and it can significantly benefit from another round of revision. However, I won't object to accepting it if my co-reviewers champion it.

**Missing References:**

I am not an expert on this task and have no more references to provide.

**Paper Topic And Main Contributions:**

This paper focuses on the financial risk detection task, which is to predict the risk label distribution of the target news piece. The proposed framework, STINMatch, is a text-graph joint modeling framework to exploit both the text information and the graph (news-company) information for better risk label diffusion. Experiments on a newly constructed dataset and two public datasets show that the proposed STINMatch outperforms baselines in four categories.

**Questions For The Authors:**

- A. In L167, the labels Y are in R^k. I wonder why the label values scatter across the real number set. And could you please give an example whose Y has a certain dimension of a very large value? In L179, the news piece in the NEKG has K binary risk labels. This is inconsistent with the former formulation.
- B. Why are the information providers anonymous? And how could we know how authoritative the information is without knowing about the source?
- C. Have you ever considered what if a piece of news indicates no risk of a related company? Is this in the research scope? If not, what pre-processing steps are required, and how to do them?

**Reasons To Accept:**

1. The task of financial risk detection is of great real-world application value.
2. The experiment part is diverse and it shows the superiority of the proposed STINMatch framework.
3. The overall writing is good. The notations, figures, and organization are carefully designed.
4. The authors acknowledge the increasing computation cost of the proposed method in the limitation section with a reasonable and clear analysis, which is seldom across my batch of submissions to be reviewed.

**Reasons To Reject:**

1. This motivation is illustrated by only listing existing works and the proposed one, lacking an analysis of the key challenges.
   - First, in L060-061, the authors say that the proposed method is unlike previous graph neural network (GNN) methods for text analysis. However, readers could not know how these previous methods do it and what is different if a method should focus on multi-label-correlation guided text-graph joint learning. Is the difference in multi-labeling, correlation, guidance, or text-graph joint learning?
   - Second, in L075-077, the case is similar. The authors state that few studies have been carried out on the semi-supervised integration for text-graph joint learning framework. I wonder what is different and challenging again because the situation that few studies consider semi-supervised integration for this framework is caused by non-technical factors (e.g., no real-world requirement, existing techniques are enough).
2. Some basic information about the data source is not disclosed. We know little about how reliable the new NEKG dataset is.
3. The task definition is confusing. Intuitively, a task named risk detection is to detect risky objects, i.e., to say whether a sample is risky or not (binary). According to Section 3, financial risk classification might be a more proper name for this task. Or a no-risk label should be included in the label space.


**Reproducibility:**

4: Could mostly reproduce the results, but there may be some variation because of sample variance or minor variations in their interpretation of the protocol or method.

**Reviewer Confidence:**

3: Pretty sure, but there's a chance I missed something. Although I have a good feel for this area in general, I did not carefully check the paper's details, e.g., the math, experimental design, or novelty.

**Typos Grammar Style And Presentation Improvements:**

Reference issues:
1. If possible, please add URLs or DOIs for all the entries. This would make it easier for readers using computers to redirect to the original papers.
2. The paper "Multi-label text classification using attention-based graph neural network" has been published at ICAART 2020. Please do not cite its preprint version.

Presentation issues (maybe it would be better to copy the main text to grammar-checking tools before submitting it):
- L041: utilizing -> utilize
- l051: The abbreviation NLP is unnecessary.
- L091: a -> an
- L239: Fig.2 -> Fig. 2
- L609: section -> Section
- L594: 10,00 -> 1,000
- L655: A ending period is missing.
- Table 1: Seems unnecessary to use "Cat" as a short name for "Category". The space is enough.

---

> ### Author Rebuttal · Authors · 2023-08-28
>
> Thank you for your insightful feedback. We are committed to improving our research study, so we eagerly look forward to incorporating your suggestions for further refinement.
>
> **Response to Question A**(*Have you ever considered what if a piece of news indicates no risk of a related company? Is this in the research scope? If not, what pre-processing steps are required, and how to do them?  *):
>
> - The scope of our research encompasses companies with varying levels of risk, including the ones with “no risk”. As outlined in Section 5.3 (enterprise risk evaluation), our dataset includes those companies that exhibit low risk, including those with no risk, as illustrated in Line 532. In response to your question, we will enhance our revision with an expanded narrative to further address this issue.
>
>
> **Response to Question B**(*Why are the information providers anonymous? And how could we know how authoritative the information is without knowing about the source?    *):
>
> - We anonymize the information providers to meet the anonymous requirement during the double-blind review period. We will provide detailed data information in the revision if the paper is accepted. This enterprise information is sourced from a government-backed open data provider,  and it enables the designed web-crawler to collect the corresponding news for annotations and evaluations. We will provide explanations in the final version.
>
> **Response to Question C**(*In L167, the labels Y are in R^k. I wonder why the label values scatter across the real number set. And could you please give an example whose Y has a certain dimension of a very large value? In L179, the news piece in the NEKG has K binary risk labels. This is inconsistent with the former formulation.  *):
>
> - We would like to clarify that in L167, $R^K$ represents $K$-dimensional vector space with real numbers and $y\in R^K$ represents that $y$ is a $K$-dimensional vector. We will make this clearer in the next version.
>
> **Response to "Typos Grammar Style And Presentation Improvements"**:
>
> - We are appreciative of the thoughtful comments, and are eager to address your concerns in the forthcoming revision of our paper. Your thoughtful feedback is essential to our continued growth. Please find our responses in the following content.
>
> **Response to "Reasons To Reject"**:
> - **Difference with previous methods**:
>     - Our work advocates for an innovative approach to financial risk detection that integrates label correlation effectively into the text-graph iterative learning process.
>     - The integration of label correlation into the iterative learning process offers a mutual benefit, significantly improving the overall performance of the system. On the one hand, the label correlation, serving as domain-specific knowledge, plays an important role to guide the label diffusion for both text and graph modules in each iteration; on the other hand, the predictions of the successively enhanced model help involve more previously unlabeled samples into the calculation of label correlation, resulting in a more generalized label correlation in each iteration round.
>     - We will rewrite this statement to make the model novelty more understandable.
>
> - **Real-world requirement of the proposed technique**:
>     - Thanks for your comments. This paper proposes leveraging a text-graph joint learning framework equipped with semi-supervised learning techniques to develop an effective multi-label financial risk detection system. To tackle the expensive data annotation cost which is traditionally required for such applications, we will make the annotation data publicly available.
>
>     - The utility of real-world applications in scenarios similar to ours is substantial, ranging from multi-label document classification based on citation or social networks, to multi-label product classification through eCommerce platforms. Nevertheless, obtaining labels for these datasets can require a lot of resources, particularly if the tasks are linked to domains such as medicine or technology that require specialized knowledge. Traditional text-graph joint learning approaches are ineffective in tackling this difficulty due to their lack of semi-supervised learning techniques.
>     - Following your suggestion, we will host additional narrative to address this problem.
>
> - **Basic information about the data source**:
>     - Please refer to the response to * Response to Question B"*.

---

### Official Review · Reviewer_oisY · 2023-08-09

**Typos Grammar Style And Presentation Improvements:** None
**Soundness:** 4

**Excitement:**

3: Ambivalent: It has merits (e.g., it reports state-of-the-art results, the idea is nice), but there are key weaknesses (e.g., it describes incremental work), and it can significantly benefit from another round of revision. However, I won't object to accepting it if my co-reviewers champion it.

**Missing References:**

None

**Paper Topic And Main Contributions:**

This paper proposes improvements for utilizing NEKG in financial risk detection. First, it proposes a semi-supervised text-graph joint learning framework that fully exploits semantic information and topological associations. Second, it introduces a content-label-topology aggregation mechanism to address multi-label diffusion issues on text-attributed graphs. Third, it releases a NEKG dataset annotated with multiple financial risks. The proposed method is evaluated on three datasets and achieves higher accuracy and F1 scores compared to baseline methods.

**Questions For The Authors:**

A.The description of existing work is unclear. For instance, the reference to "two modules" in line 136 and "single module" in line 159 is ambiguous.
B.Line 231-line 233: Why can the diffusion model provide the probability distribution of risk labels for enterprises? What prevents other methods from achieving the same result?
C.The purpose of data enhancement is only to calculate supervised loss and unsupervised loss. Why did the authors employ both strong and weak enhancements?
D.Calculating the similarity between different labels can also measure their associations. What are the advantages of the author's proposed association matrix R?
E.The description of the baseline methods is unclear. Which baseline methods are considered multi-label methods? How was the conclusion in lines 508-510 derived?
F.Comparing performance with the simplest TCB in the time cost section is not convincing.
G.The generalization experiments on the two public datasets do not include all baseline methods. Why?

**Reasons To Accept:**

1.The proposed method performs well even when the labeled data size is small.
2.It reduces dependency on labeled data, resulting in significant cost savings in terms of annotation.
3.The method achieves a 4% improvement in Macro-F1 for the news risk detection task.

**Reasons To Reject:**

1.The writing of the paper is not satisfactory, and has numerous and chaotic symbols in the text. A symbol table and an algorithmic table would make the paper clearer.
2.The rationale behind the use of weak augmentation and strong augmentation is not thoroughly explained.
3.The effectiveness of joint learning has not been evaluated.

**Reproducibility:**

4: Could mostly reproduce the results, but there may be some variation because of sample variance or minor variations in their interpretation of the protocol or method.

**Reviewer Confidence:**

4: Quite sure. I tried to check the important points carefully. It's unlikely, though conceivable, that I missed something that should affect my ratings.

---

> ### Author Rebuttal · Authors · 2023-08-28
>
> We are immensely grateful for your thoughtful review of our paper, and for taking the time to provide vital feedback. Our team has carefully weighed the doubts you raised and we are providing the following response.
>
> **Response to Question A** (*Calculating the similarity between different labels can also measure their associations. What are the advantages of the author's proposed association matrix R?*):
> - The proposed correlation matrix R exactly measures the similarity between different labels, as indicated in Line 377-380: *“we utilize the correlation matrix R to capture the internal relations among different labels to enable cross-label similarity calculation”*. More specifically, the matrix measures the similarity between the distribution of risk labels among samples.
>
> - By leveraging the proposed STINMatch method, we seek to optimize the integration of the correlation matrix R into an iterative joint learning framework in order to maximize performance and efficiency. Our approach aims to refine the calculation of the correlation matrix R, enabling the iterative joint learning framework to reach its full potential while still maintaining optimal levels of efficiency. We appreciate your question, and we will provide additional narrative in the revision to highlight this problem.
>
>
> **Response to Question B** (*	Comparing performance with the simplest TCB in the time cost section is not convincing.*):
>
> - The ”Time Cost Analysis“ serves to assess the time performance of the proposed method, which is a potential limitation of the proposed method based on joint learning. The additional time cost can be introduced by the graph-text iterative learning framework as well as the semi-supervised techniques, which involve more training rounds and data points in the learning process. To evaluate the worst case of this limitation, we compare the time cost between STINMatch and the simplest baseline, TCB, which takes the least training time among all baselines. Our results indicate that the additional time cost introduced by STINMatch is tolerable regarding the improvement in model utility (e.g., enhancing 9.5% F1).
>
> **Response to Question C** (*Line 231-line 233: Why can the diffusion model provide the probability distribution of risk labels for enterprises? What prevents other methods from achieving the same result?*):
>
> - The proposed graph module of the diffusion model is effective in propagating risk labels from news to enterprises during its training process. Unlike other baseline models, this diffusion model reflects a real-world risk diffusion process, wherein the risk labels for enterprises are randomized at the outset. Upon the diffusion model reaching convergence, the enterprise risk labels are adopted as signals for precisely quantifying risk evaluation. The proposed semantic-topological-iteration training framework and the accompanying semi-supervised learning mechanism signify an advancement in how we propagate risk labels in graph-based methods, which differs from other baselines. With our propositions, we see an increase in the effectiveness of the process. We will host additional content in the revision to address this problem.
>
> **Response to Question D** (*The generalization experiments on the two public datasets do not include all baseline methods. Why?  *):
>
> - Thank you for this important comment. Unfortunately, because of the space limitation, we cannot host all the experiment outcomes in the paper. Here, we supplement the results for other baselines in the below table:
>
> | Method           | RentTheRunWay |         | Goodreads-Spoilers |         |
> |------------------|---------------|---------|--------------------|---------|
> |                  | MacroF1       | MicroF1 | MacroF1            | MicroF1 |
> |FastText         | 0.448         | 0.727   | 0.551              | 0.581   |
> |TextRCNN$^\dagger$         | 0.467         | 0.748   | 0.563              | 0.601   |
> |TextRNN-attention$^\dagger$ | 0.465         | 0.746   | 0.566              | 0.604   |
> |TextCNN$^\dagger$          | 0.469         | 0.749   | 0.564              | 0.603   |
> |RoBERTa| 0.481         | 0.764   | 0.573              | 0.613   |
> |LCNNI$^\dagger$            | 0.471         | 0.752   | 0.560              | 0.592   |
> |SGM              | 0.463         | 0.746   | 0.561              | 0.594   |
> |CORE$^\dagger$             | 0.486         | 0.771   | 0.576              | 0.615   |
> |TCB-Focal-loss   | 0.476         | 0.761   | 0.565              | 0.605   |
> |TCB-DB-loss      | 0.475         | 0.760   | 0.564              | 0.605   |
> |Pseudo-TCB       | 0.474         | 0.758   | 0.567              | 0.607   |
> |MixMatch-TCB     | 0.484         | 0.766   | 0.573              | 0.612   |
> |ReMixMatch-TCB   | 0.483         | 0.764   | 0.571              | 0.610   |
> |FixMatch-TCB| 0.495         | 0.786   | 0.584              | 0.621   |
> |GCN-TCB          | 0.481         | 0.765   | 0.571              | 0.609   |
> |GraphSAGE-TCB    | 0.484         | 0.767   | 0.573              | 0.612   |
> |GAT-TCB| 0.487         | 0.771   | 0.579              | 0.613   |
> |MAGNET-TCB       | 0.485         | 0.769   | 0.574              | 0.609   |
> |LC-GAT-TCB       | 0.496         | 0.782   | 0.581              | 0.620   |
> |GLEM-TCB| 0.501         | 0.789   | 0.585              | 0.624   |
> | SSINMatch        | 0.516         | 0.801   | 0.615              | 0.638   |
>
> - If space permits, we will include the below table in the revised manuscript. Otherwise, this information will be readily accessible on the project website.
>
>
>
>
> **Response to Question E** (*The description of existing work is unclear. For instance, the reference to "two modules" in line 136 and "single module" in line 159 is ambiguous. *):
>
> - Our sincerest gratitude is extended for your detailed suggestions on this work. We will certainly incorporate your critical insights to strengthen the content. The "two modules" in line 136 refer to a graph module and a text module, as indicated in line 131: *‘Some recent works are combining GNNs with text classifiers to take advantage of both…”*.
> The “single module” in line 159 means a single text module or a single graph module.
> Since these descriptions are not direct enough, we will clarify this in the next version. We will rewrite this section.
>
>
>
> **Response to Question F** (*The purpose of data enhancement is only to calculate supervised loss and unsupervised loss. Why did the authors employ both strong and weak enhancements?  *):
>
> - Please refer to the response to *Explanation to strong and weak augmentation* in * Response to "Reasons To Reject"*.
>
>
> **Response to Question G** (*The description of the baseline methods is unclear. Which baseline methods are considered multi-label methods? How was the conclusion in lines 508-510 derived?*):
>
> - The *multi-label methods* in line 509 refer to Category 2 baselines in Section 5.2, which are state-of-the-art text models specialized for MLTC tasks (see L483) where MLTC refers to multi-label text classification (see L51).
>
> - From the results in Cat2 of Table1, we can derive the conclusion in lines 508-510: “One can witness slight improvements with some multi-label methods, indicating the usefulness of correlations among labels”. We will make this paragraph clearer in the next version.
>
>
> **Response to "Reasons To Reject"**:
>
>
> - **Evaluation on the effectiveness of joint learning**:
>     - We apologize for any confusion the evaluation section may have caused and seek to clarify the settings and motivation behind each comparison experiment in the revision.
>     - In Section 5.3, we evaluated the effectiveness of joint learning by comparing the results between the proposed method and the baselines in Category 2, which are semi-supervised-enhanced text classification without text-graph joint learning. We will point this out to clarify that baselines in Category 2 are involved for evaluating the improvement by text-graph joint learning.
>     - In addition, the two descriptions "w/o unsupervised loss for $M_g$" and "w/o unsupervised loss for $M_t$" in the ablation study in Section 5.4 indicate the results after removing the semi-supervised techniques from the text model and the graph model, respectively. This serves to evaluate the effectiveness of the semi-supervised component. We will rewrite this part.
>
>
>
> - **Explanation to strong and weak augmentation**:
>    - We employ weak and strong data augmentation following the intuition of consistency regularization as described in Line 151-153: “use weakly-augmented examples to generate artificial labels and enforce consistency against strongly augmented examples.” More specifically, the goal of consistency regularization is to encourage the model to produce similar predictions for similar inputs even under different perturbations. Some previous studies, such as ReMixMatch and FixMatch, have shown the effectiveness of using both weak and strong augmentations for consistency regularization in semi-supervised learning. Similarly, we use the pseudo labels obtained from weak augmentation to fit the samples with strong augmentation, so that the model can better adapt to unseen data. The weak augmentation provides higher accuracy for the pseudo labels, while strong augmentation provides better diversity and a larger region of sample perturbation for the consistency regularization, thereby improving the performance of the semi-supervised learning.
>
>     - We will host additional narrative in the revision to address your concern and also help readers to understand the difference between strong and weak augmentations.
>
>
> - **Notation Table**:
>
>      - We appreciate your important input and concur that the addition of an annotation table would strengthen the paper and make it easier to understand. To that end, we will incorporate the following table into our revision to provide clarity for all symbols.
>
>     | Symbol     | Description                                                   |
>     | ----------- |---------------------------------------------------------------|
>     |$\mathbb R$| label correlation matrix                                      |
>     |$K$| 	number of all risk labels                                    |
>     |$N$| vertex set for news                                           |
>     |$C$| vertex set for enterprises                                    |
>     |$E$| edge set of news-enterprise KG                                |
>     |$D_L$| labeled news data                                             |
>     |$D_U$| unlabeled news data                                           |
>     |$M_g$| graph module                                                  |
>     |$M_t$| text module                                                   |
>     |$x_t$| output of the text encoder in $M_t$                           |
>     |$\psi_t$| hidden representation in $M_t$                                |
>     |$l_t$| output of $M_t$                                               |
>     |$l_t^s$| predictions on strong augmentation data in $M_t$              |
>     |$l_t^w$| predictions on weak augmentation data in $M_t$                |
>     |$h_g$| node representation in $M_g$                                  |
>     |$\psi_g$| hidden representation in $M_g$                                |
>     |$l_g$| output of $M_g$                                               |
>     |$l_g^s$| predictions on strong augmentation data in $M_g$              |
>     |$l_g^w$| predictions on weak augmentation data in $M_g$                |
>     |$\beta$| node-level attention weights regarding label correlation      |
>     |$\eta$| similarity-based node-level multi-head attention weights      |
>     |$\alpha$| node-level attention weights in $M_g$                         |
>     |$\gamma$| balance weight parameter for unsupervised and supervised loss |

---

### Official Review · Reviewer_zFLA · 2023-08-12

**Soundness:** 4

**Excitement:**

4: Strong: This paper deepens the understanding of some phenomenon or lowers the barriers to an existing research direction.

**Paper Topic And Main Contributions:**

The paper focuses on detecting risks associated with information from news (10 risks in particular). The proposed method (STINMatch) is a semi-supervised learning framework that uses semantic information and topological association (via a News-Enterprise Knowledge Graph aka NEKG) to evaluate the propagation of news risk and enterprise risk. Along with the proposed framework, the paper also provides a NEKG dataset annotated with financial risks, as well as a new aggregation mechanism is proposed to handle multi-label propagation on text-attributed graphs.

**Questions For The Authors:**

Question A: Are risks somewhat uniformly distributed? Or is there a particular risk that is more common? Do most news have a risk associated with them, or are they mostly neutral (highly imbalanced dataset)?

**Reasons To Accept:**

Very thorough work is done to evaluate the effectiveness of STINMatch as well as in investigating the contribution of each component in the Ablation Study.

The creation of the NEKG dataset can be very influential in the financial domain.

The methodological contribution of STINMatch is provided quite clearly and coherently.

**Reasons To Reject:**

No reason to reject. However, for the Enterprise Risk Evaluation, the decision to divide credit risk labels between high-risk and low-risk based on having more than one predicted risk label does not have strong reasoning. Since information regarding the distribution of risks (see Question A) is missing, whether or not this is a good criterion for classifying an enterprise as high or low risk is unclear. Risks such as Bankruptcy risk may pose a greater financial risk than others for example.

**Reproducibility:**

4: Could mostly reproduce the results, but there may be some variation because of sample variance or minor variations in their interpretation of the protocol or method.

**Reviewer Confidence:**

3: Pretty sure, but there's a chance I missed something. Although I have a good feel for this area in general, I did not carefully check the paper's details, e.g., the math, experimental design, or novelty.

---

> ### Author Rebuttal · Authors · 2023-08-28
>
> We extend our sincerest appreciation for you taking the time to review our paper and providing us with in-depth feedback. We are delighted that you recognize the novelty and potential of our idea. After careful consideration of your questions and concerns, we have formulated these feedbacks to address them.
>
>
> **Response to Question A** (*Are risks somewhat uniformly distributed? Or is there a particular risk that is more common? Do most news have a risk associated with them, or are they mostly neutral (highly imbalanced dataset)?*):
>
> - Following your suggestion, we made a table to show the risk distribution among all 15,000 annotated news, which shows that Litigation, Bankruptcy and Dispute related risks can be more popular than others.
>
>     |**Risk Label**    |**Number of news**|
>     | ----------- | ----------- |
>     |Bankruptcy|	3376|
>     |Liquidation|2107|
>     |Business closure|	272|
>     |Production halts|	705|
>     |Debt|	4097|
>     |Corruption|	474|
>     |Dispute|	5262|
>     |Counterfeit|	1187|
>     |Fraud	|993|
>     |Litigation|	8277|
>
> - Meanwhile, we report the circumstances of multi-label annotation of the news data. Among all 15,000 annotated news, 5855 news are associated with 1 risk label, 6986 news are associated with 2 risk labels, and more than two thousand news have 3 or more labels.
>
>     |**Risk Label Count**|**Number of news**|
>     |---------------------| ----------- |
>     | 1                   |	5855|
>     | 2                   |	6986|
>     | 3                   |	1725|
>     | 4                   |	344|
>     | 5                   |	83|
>     | 6                   |	7|
>
> We will supplement this information in the next version. A more detailed data description will be available on the project website.

---

### Meta-Review · Area_Chair_3XLY · 2023-09-15

**Recommendation:** 4

**Metareview:**

The proposed method (STINMatch) is a semi-supervised learning framework that uses semantic information and topological association (via a News-Enterprise Knowledge Graph aka NEKG) to evaluate the propagation of news risk and enterprise risk.

This approach is really interesting although some concerns are mainly related:
1) The task definition is not properly defined; specifically more relevant definitions may support the readers in understanding the task
2) Some basic information about the data source is not disclosed.
3) Key challenges need to be more properly defined
4) More clarifications about the framework need to be provided (also improving Figure 1).

---

### Decision · Program_Chairs · 2023-10-07

**Decision:**

Accept-Main

**Comment:**

The proposed method (STINMatch) is a semi-supervised learning framework that uses semantic information and topological association (via a News-Enterprise Knowledge Graph aka NEKG) to evaluate the propagation of news risk and enterprise risk.

This approach is really interesting although some concerns are mainly related:
1) The task definition is not properly defined; specifically more relevant definitions may support the readers in understanding the task
2) Some basic information about the data source is not disclosed.
3) Key challenges need to be more properly defined
4) More clarifications about the framework need to be provided (also improving Figure 1).